# One-D-Piece: Image Tokenizer Meets Quality-Controllable Compression

**Keita Miwa** [1]  **Kento Sasaki** [1]  **Hidehisa Arai** [1]  **Tsubasa Takahashi** [1]  **Yu Yamaguchi** [1]

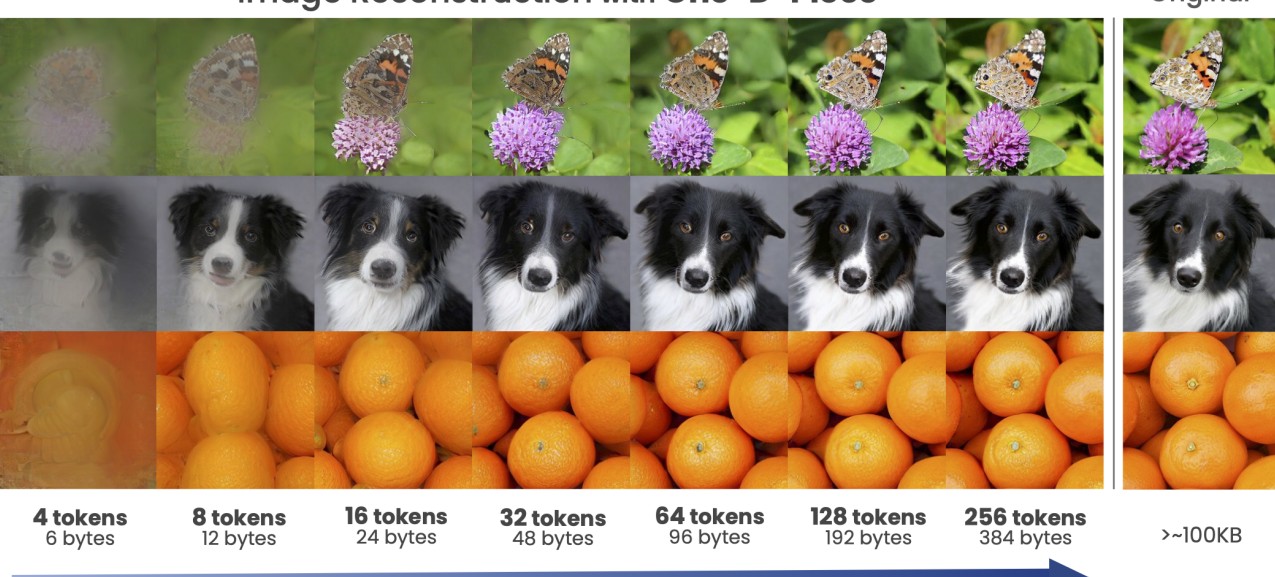

Figure 1: We propose **One-D-Piece**, discrete image tokenizer that enables variable-length tokenization adjustable from 1 to 256 tokens. Even with a very small number of tokens (e.g., $n_{\text{tokens}} = 8$), it achieves recognizable image reconstructions. As the token count increases, the image quality progressively improves, reaching near-original fidelity at $n_{\text{tokens}} = 256$.

## Abstract

Current image tokenization methods require a large number of tokens to capture the information contained within images. Although the amount of information varies across images, most image tokenizers only support fixed-length tokenization, leading to inefficiency in token allocation. In this study, we introduce One-D-Piece, a discrete image tokenizer designed for variable-length tokenization, achieving quality-controllable mechanism. To enable variable compression rate, we introduce a simple but effective regularization mechanism named "Tail Token Drop" into discrete one-dimensional image tokenizers. This

method encourages critical information to concentrate at the head of the token sequence, enabling support of variadic tokenization, while preserving state-of-the-art reconstruction quality. We evaluate our tokenizer across multiple reconstruction quality metrics and find that it delivers significantly better perceptual quality than existing quality-controllable compression methods, including JPEG and WebP, at smaller byte sizes. Furthermore, we assess our tokenizer on various downstream computer vision tasks, including image classification, object detection, semantic segmentation, and depth estimation, confirming its adaptability to numerous applications compared to other variable-rate methods. Our approach demonstrates the versatility of variable-length discrete image tokenization, establishing a new paradigm in both compression efficiency and reconstruction performance. Finally, we validate the effectiveness of tail token drop via de-

---

[1]Turing Inc., Tokyo, Japan. Correspondence to: Keita Miwa <miwakeita@turing-motors.com>.

*Proceedings of the ICML 2025 Tokenization Workshop (TokShop)*, Vancouver, Canada. PMLR 267, 2025. Copyright 2025 by the author(s).

tailed analysis of tokenizers. Our Project Page is at https://turingmotors.github.io/one-d-piece-tokenizer/.

## 1. Introduction

In recent years, with the rapid advancements in vision-language models (VLMs) (Liu et al., 2024; Bai et al., 2023; Laurençon et al., 2024) and image and video generation models (Yan et al., 2021; Villegas et al., 2022; Bruce et al., 2024), the concept of discrete tokenization for visual data, similar to language tokenization, has garnered increasing attention (van den Oord et al., 2017; Yu et al., 2024a; Luo et al., 2024; Wang et al., 2024a). This approach enables seamless integration with Transformer-based models (Vaswani et al., 2017), simplifying model architectures and reducing computational complexity.

However, challenges remain in discrete tokenization for visual data, particularly in capturing spatial structures, which often requires long, fixed-length token sequences. For instance, typical image tokenizers require as much as 256 tokens to represent a single 256×256 pixel image, which limits their flexibility in practical applications. To overcome this limitation, one-dimensional (1D) tokenization methods have emerged, aiming to achieve higher compression rates while maintaining reconstruction quality. Notably, the SEED tokenizer (Ge et al., 2024a) can semantically represent images by causal 1D sequences, while the TiTok tokenizer (Yu et al., 2024b) achieves high-quality image reconstruction with just 32 tokens. These approaches efficiently encode the entire image into a compact 1D sequence, significantly reducing the number of tokens required. Nevertheless, there exists a fundamental trade-off between compression rate and reconstruction quality (Shannon, 1959; Blau & Michaeli, 2019); higher compression results in greater degradation, especially for complex images. As current image tokenizers are designed to produce fixed number of tokens, it is impossible to control the quality based on specific requirements.

In contrast, classical image compression methods such as JPEG (Wallace, 1991) have long addressed this trade-off by allowing users to adjust compression rates based on the desired quality. These methods provide well-established mechanisms for balancing file size and visual fidelity, making them highly versatile across various applications. However, traditional compression algorithms are not designed for direct use as input representations for neural networks, making it challenging to integrate them into neural models like VLMs. Furthermore, these algorithms differ fundamentally from the adaptive, model-driven strategies used in modern image tokenizers, complicating the transfer of established compression techniques to the tokenization domain.

Given these differences, there is a pressing need for a novel

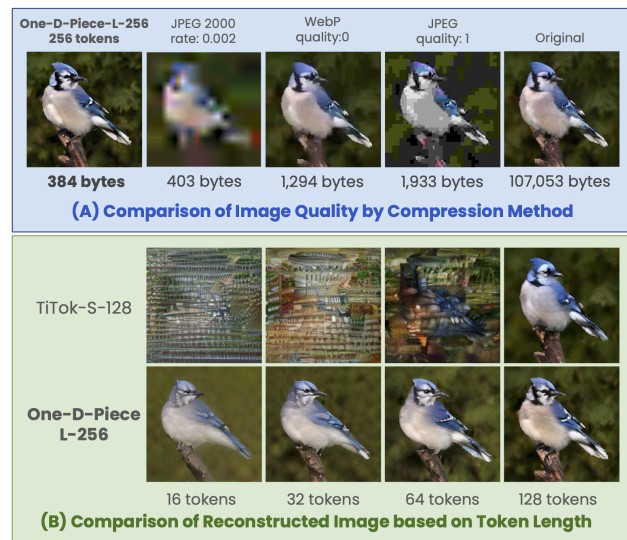

Figure 2: **Comparison of Image Quality and Compression Efficiency.** (A) One-D-Piece-L-256 achieves superior visual quality with enhanced compression efficiency, reducing image size to 384 bytes, outperforming image formats. (B) One-D-Piece-L-256 improves reconstruction quality as token length increases, proving effective with fewer tokens (e.g., 16, 32, 64, and 128). In contrast, TiTok-S-128, trained with a fixed token length of 128, can reconstruct images only when all 128 tokens are used.

approach that bridges the adaptability of tokenization methods with the efficiency of traditional compression formats. To address this challenge, we propose *One-D-Piece*, a new variable-length discrete image tokenizer that combines the benefits of tokenization with the flexibility of classical compression methods. Our approach introduces a simple yet effective regularization technique, *Tail Token Drop*, which concentrates critical information at the beginning of the token sequence, enabling efficient and adaptive token lengths ranging from 1 to 256 tokens. This allows for maintaining high reconstruction quality even with a low token count, providing flexible compression suited for diverse applications. As illustrated in Figure 1 and Figure 2, our model supports variable-length image tokenization and can effectively produce visually accurate tokenizations and reconstructions even with as few as 8 or 16 tokens.

We further evaluate our method not only using standard image quality metrics but also by assessing its performance on a range of downstream tasks, demonstrating practical benefits over classical image compression techniques. We showcase the effectiveness of One-D-Piece through extensive experiments on various computer vision tasks, including image classification, object detection, and semantic segmentation. Our results indicate that One-D-Piece outperforms existing variable-length compression methods, including JPEG (Wal-

lace, 1991), JPEG 2000 (iso, 2019), and WebP (Google, 2024), in terms of perceptual quality, especially at low token counts.

Unlike existing variable quality control methods, we focus on ultra-low bitrate image tokenizers, enabling high compression with superior perceptual quality. ElasticTok (Yan et al., 2024) is the closest work to ours, but it targets a completely different bitrate regime. A detailed comparison with prior works is provided in Section 2.1.

Our main contributions can be summarized as follows:

- We introduce One-D-Piece, a variable-length discrete image tokenizer utilizing a novel Tail Token Drop regularization technique.

- Our method achieves competitive reconstruction quality while supporting flexible token lengths and is validated through comprehensive evaluations, including downstream tasks, demonstrating superiority over traditional image compression methods.

- We analyze our models' behaviors in detail, revealing that the Tail Token Drop method effectively concentrates important information at the head and further provides insights into image generation and adaptive token allocation.

## 2. Related Work

### 2.1. High-Compression Image Tokenization

Pixel-space image representations are inefficient for image generation and vision-language tasks. To overcome this, models like VAEs (Kingma & Welling, 2014) extract latent features, while encoders such as CLIP (Radford et al., 2021) capture compact semantic representations. More recently, discrete image tokenization methods, such as VQ-VAE (van den Oord et al., 2017), have gained traction for compressing images into discrete tokens, facilitating both image generation (Esser et al., 2021; Chang et al., 2022; Weber et al., 2024; Yan et al., 2021; Villegas et al., 2022; Bruce et al., 2024) and integration with text tokens (Team, 2024; Wang et al., 2024b; Ge et al., 2024b).

While "2D tokenizers," which utilize CNNs as encoders and decoders and maintain a two-dimensional structural representation of the image, are typically studied (van den Oord et al., 2017; Yu et al., 2024a; Luo et al., 2024; Wang et al., 2024a; Esser et al., 2021; Weber et al., 2024; Chang et al., 2022), they often achieve low compression efficiency due to their limited ability to effectively capture global information. Typical 2D tokenizers consume 256 tokens for 256×256 images. In contrast, recent advances have introduced "1D tokenizers" that compress images by capturing both local and global information simultaneously, using architectures

like Transformers to process the entire image context. For instance, SEED-Tokenizer (Ge et al., 2024a) uses Vision Transformer (ViT) (Dosovitskiy et al., 2021) and reduces an image to a sequence of semantic tokens, demonstrating efficient compression for use in VLMs (Ge et al., 2024b; Wang et al., 2024b). Another high-compression 1D tokenizer, TiTok (Yu et al., 2024b), effectively compresses images into just 32 sequential tokens while preserving high visual fidelity, outperforming existing 2D tokenizers in both compression efficiency and reconstruction quality.

### 2.2. Variable-Rate Image Representation

In the field of image codecs where the primary motivation is pure compression rather than generation or vision-language integration, it is well known that high compression is not a free lunch (Shannon, 1959; Blau & Michaeli, 2019). There exists a trade-off between rate and reconstruction quality, which becomes more severe for complex images.

Rate-distortion trade-offs are mitigated by variable-rate compression, which allows greater flexibility based on purpose. Standard codecs like JPEG (Wallace, 1991), JPEG 2000 (iso, 2019), and WebP (Google, 2024) achieve this through transform and entropy coding, such as the Discrete Cosine Transform and Huffman coding used in JPEG.

Variable quality control has also been explored. Matryoshka Representation Learning (MRL) (Kusupati et al., 2022) is used for variable-rate representation learning. By applying losses only to fixed $k$ lengths, specifically at $k \approx \lfloor \log n \rfloor$, MRL has been experimentally shown to generalize across all possible lengths. A simpler approach, "Tail Drop" (Koike-Akino & Wang, 2020) concentrates critical information towards the *head* of the embeddings. This method is applied in AutoEncoder models by imposing a higher dropout rate on the *tail* of the latent representation. By focusing essential information towards the beginning of the latent sequence, Tail Drop enhances compression efficiency, enabling variable-rate compression. While MRL requires $\mathcal{O}(\log n)$ computational cost for loss calculation, Tail Drop requires only $\mathcal{O}(1)$. There are also recent studies working on variable-length discrete visual tokenization. ElasticTok (Yan et al., 2024) performs masking-based regularization for video tokenization, which is similar to Tail Drop. While they achieved high-quality rate-distortion trade-off, they lose perceptual quality at ultra-low bitrate. ALIT (Duggal et al., 2025) introduces a variable-length image tokenizer based on a recurrent allocation mechanism. While they achieved adaptive token allocation, their approach requires significantly larger inference-time cost.

In this work, we extend Tail Drop approach into ultra-low bitrate image tokenizers, which enables high compression with superior perceptional quality.

# 3. Method

We introduce **One**-**D**imensional Image **Piece** Tokenizer (One-D-Piece), a novel discrete tokenizer designed for efficient image compression. In contrast to existing image tokenizers that typically generate fixed-length sequences, One-D-Piece produces variable-length tokens, similar to text tokenizers like WordPiece (Wu et al., 2016) and SentencePiece (Kudo & Richardson, 2018).

## 3.1. Tail Token Drop

A regularization technique "tail drop" was originally introduced to improve compression efficiency by dynamically controlling the dimensionality of the latent representation (Koike-Akino & Wang, 2020). Tail drop can be viewed as a variant of Dropout, where the dropout rate progressively increases towards the end (tail) of the latent vector. This technique prioritizes learning the most essential features in the earlier (head) neurons, while gradually discarding less critical information in the tail. As a result, tail drop enables flexible compression, where the dimensionality can be adjusted at inference time with controlling the quality of the reconstruction.

We adapt this simple yet effective method for image tokenization with modifications, introducing "Tail Token Drop," which involves randomly truncating the tail of the token sequence and can be applied to 1D tokenizers, which produces structure-free image tokens.

Formally, let $\mathbf{q} = [q_1, q_2, \ldots, q_N]$ be the token sequence generated by our 1D tokenizer, where $q_i$ represents each token in the sequence, and $N$ is the total number of tokens. During training, the number of tokens to be dropped, $k$, is sampled from a uniform distribution: $k \sim U(0, N-1)$.

Thus, the token sequence after applying Tail Token Drop, denoted as $\mathbf{q}'$, is given by: $\mathbf{q}' = [q_1, q_2, \ldots, q_{N-k}]$.

By applying this regularization, the tokenizer is encouraged to accumulate more critical information towards the beginning of the sequence by randomly truncating tail tokens and less significant information tends to accumulate at the tail, which is more likely to be truncated. As a result, the tokenizer trained with Tail Token Drop allows the token sequence length to be flexibly adjusted based on the information content of the image, by cutting the tails adaptively.

## 3.2. Architecture

The architecture of One-D-Piece focuses on two important requirements. First, the tokens produced must form a 1D sequence. They should not explicitly correspond to 2D structure, like most 2D tokenizers. This restriction exists because the tail cannot be defined for 2D tokens, hindering application of Tail Token Drop. Second, the detokenizer

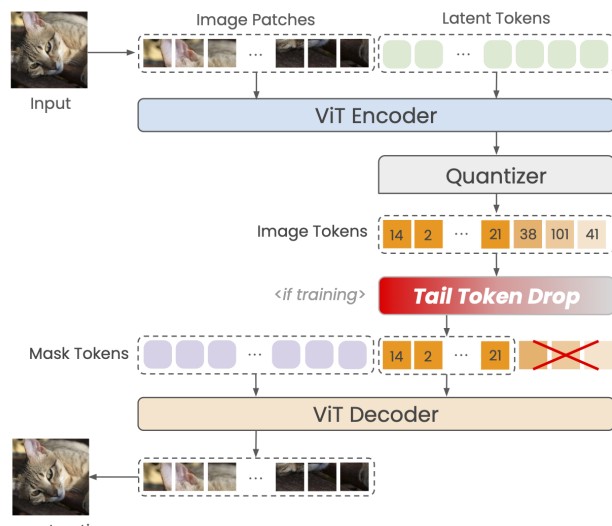

Figure 3: One-D-Piece applies random Tail Token Drop during training to concentrate the most important information at the head of the token sequence.

must handle variable-length of tokens; otherwise, the Tail Token Drop technique cannot be applied.

To meet these requirements, we build upon the TiTok architecture for One-D-Piece (Figure 3). The TiTok model (Yu et al., 2024b) consists of three main components: the encoder, quantizer, and decoder. The encoder first divides the input image $\mathbf{X} \in \mathbb{R}^{H \times W \times C}$ patches. Each patch is embedded into a vector, yielding a set of patch embeddings. We also include $N$ learnable latent tokens. The output of the ViT encoder, $\mathbf{z}$, corresponding to the latent tokens is used as latent space. The output $\mathbf{z}$ is then discretized by the quantizer, producing a discrete set of token representations $\mathbf{q} = \text{Quantizer}(\mathbf{z})$. This quantized sequence serves as a discrete representation of the image. For One-D-Piece training, we apply Tail Token Drop regularization to this as $\mathbf{q}' = \text{TailTokenDrop}(\mathbf{q})$. In the decoder, the quantized tokens $\mathbf{q}$ or $\mathbf{q}'$ are processed alongside mask tokens. The output corresponding to the mask tokens is upscaled by a CNN to generate the final reconstructed image $\hat{\mathbf{X}} \in \mathbb{R}^{H \times W \times C}$. An additional benefit of adopting the TiTok architecture is its flexibility in controlling the length of latent tokens as a training-time hyperparameter. Unlike simple ViT-based architectures, which constrain the token length to the number of input patches, the TiTok architecture imposes no such restriction.

## 3.3. Training

We use the two-stage training strategy adopted by TiTok. In the first stage, the model is trained to predict the logits of a pretrained tokenizer using cross-entropy loss. The second stage involves training the model to reconstruct the image

Table 1: **Reconstruction Quality across Tokenizers and Image Formats**. Other than algorithm-based image formats, One-D-Piece is the only model which support variable-length image tokenization. For these formats, byte per Image values represent can vary based on image content and quality settings.

| Method | Mechanism | Token Counts | Bits per Pixel | rFID↓ | PSNR↑ |
|---|---|---|---|---|---|
| VQGAN (Esser et al., 2021) | | 256 | 0.039 | 7.94 | 19.4 |
| MaskGIT (Chang et al., 2022) | CNN-based | 256 | 0.039 | 2.28 | — |
| LlamaGen (Sun et al., 2024) | 2D Discrete Tokenizer | 256 | 0.055 | 2.19 | 20.79 |
| VQGAN+ (Weber et al., 2024) | | 256 | 0.047 | 1.61 | — |
| Open-MAGVIT2 (Luo et al., 2024) | | 256 | 0.070 | 1.17 | 21.90 |
| JPEG (Wallace, 1991) | Algorithm-based | — | 0.252 | 113.3 | 20.99 |
| JPEG 2000 (iso, 2019) | Image Formats | — | 0.050 | 299.4 | 19.19 |
| WebP (Google, 2024) | | — | 0.240 | 31.98 | **26.18** |
| ALIT-S (IN1K) (Duggal et al., 2025) | Variable-Length | 32 to 256 | 0.0077 to 0.047 | 2.21@256 | — |
| ElasticTok-FSQ (Yan et al., 2024) | 1D Discrete Tokenizer | 256 to 4096 | 0.062 to 0.99 | 35.9@256 | 23.29@256 |
| TiTok-S-128 | ViT-based | 128 | 0.023 | 1.70@128 | 17.80@128 |
| TiTok-B-64 | 1D Discrete Tokenizer | 64 | 0.012 | 1.71@64 | 17.13@64 |
| TiTok-L-32 | | 32 | 0.0059 | 2.21@32 | 15.96@32 |
| One-D-Piece-S-256 | ViT-based | | | 1.48@256 | 18.28@256 |
| One-D-Piece-B-256 | 1D Discrete Tokenizer | 1 to 256 | 0.00018 to 0.047 | **1.11**@256 | 18.77@256 |
| One-D-Piece-L-256 | with Tail Token Drop | | | **1.08**@256 | 19.04@256 |

itself after learning the logits in the first stage. Here, the pretrained tokenizer is incorporated into the decoder, while the encoder remains frozen. The model is then optimized using reconstruction loss. The loss function includes L2 loss to reduce distortion, and perceptual loss and GAN loss for improved visual quality: $\mathcal{L}_{stage2} = \mathcal{L}_{L2} + \mathcal{L}_{Perceptual} + \mathcal{L}_{GAN}$. To support variable-length tokenization, we apply Tail Token Drop during training to dynamically adjust token lengths. For each batch, an index from 1 to 256 (representing minimum to maximum) is uniformly sampled, and tokens beyond this index are truncated.

Training and evaluation are both conducted using the ImageNet-1K dataset (Deng et al., 2009), which contains 1,000 object classes, 1,281,167 training images, 50,000 validation images, and 100,000 test images.

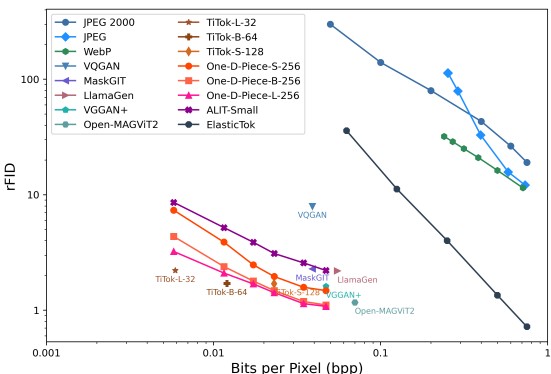

Figure 4: **Comparison of rFID by bits per pixel (bpp)**. One-D-Piece improves rFID as token length increases and achieves better scores at lower token counts than standard image formats.

## 4. Experiments

The primary goal of these experiments is to evaluate the effectiveness of our One-D-Piece across different settings, including reconstruction quality, as well as various downstream tasks. By comparing against compression algorithms including JPEG, JPEG 2000, and WebP and image tokenizers, we demonstrate the advantages of our approach.

We train One-D-Piece models using a maximum of 256 tokens. To explore different model complexities, we trained three variants, S-256, B-256, and L-256, which differ in the parameter size of the Vision Transformer. The hyperparameter settings strictly followed those used for TiTok.

Training was conducted in two phases: 100 epochs on ImageNet-1K in Stage 1, followed by 200 epochs in Stage 2. We utilize 8 NVIDIA H100 80 GB GPU. Training times vary based on model complexity: S-256 takes approximately 5 days, B-256 requires about 6 days, and L-256 takes around 9 days. This difference reflects the increased parameter count and computational demands of larger models.

### 4.1. Reconstruction

We evaluate the reconstruction quality of three variants of the One-D-Piece models and compare them to other image tokenizers and standard image formats.

Table 1 includes our main metrics, Fréchet Inception Distance (FID) (Heusel et al., 2017) for assessing perceptual quality, and PSNR for measuring distortion. Our models exhibit sufficiently better performance compared to other image tokenizers, with the B-256 and L-256 variants achieving the lowest rFID scores of 1.11 and 1.08 at 256 tokens.

For image formats, we controlled the quality settings to align

Table 2: **Evaluation across multiple downstream tasks at different token lengths compared to compression formats**. The scores in the "Base" for Image Classification and Semantic Segmentation represent comparisons between the dataset's ground truth and the model's predictions. In contrast, for Object Detection, Depth Estimation, and CLIP Emb Reconstruction, the model's predictions are used as ground truth, as these tasks do not include them in the dataset. Blue background indicates scores in which One-D-Piece-L-256 surpasses WebP.

| Task | Metrics | One-D-Piece-L-256 | | | | | Image Formats | | Base |
| | | @16 | @32 | @64 | @128 | @256 | JPEG | WebP | |
|---|---|---|---|---|---|---|---|---|---|
| Object Detection | mAP@0.5:0.95↑ | 0.051 | 0.097 | 0.180 | 0.264 | 0.305 | 0.001 | 0.166 | — |
| | mAP@0.5↑ | 0.093 | 0.163 | 0.277 | 0.377 | 0.422 | 0.001 | 0.217 | — |
| | mAP@0.75↑ | 0.049 | 0.097 | 0.185 | 0.280 | 0.323 | 0.001 | 0.178 | — |
| Depth Estimation | L1 Loss↓ | 2.436 | 1.949 | 1.480 | 1.120 | 0.965 | 3.742 | 1.553 | — |
| | L2 Loss↓ | 12.282 | 8.593 | 5.326 | 3.160 | 2.362 | 24.097 | 5.050 | — |
| CLIP Emb Reconstruction | Cos Sim↑ | 0.820 | 0.866 | 0.904 | 0.930 | 0.942 | 0.610 | 0.826 | — |
| Image Classification | Acc@1↑ | 0.504 | 0.623 | 0.731 | 0.779 | 0.792 | 0.284 | 0.664 | 0.841 |
| | Acc@5↑ | 0.718 | 0.831 | 0.908 | 0.937 | 0.946 | 0.479 | 0.870 | 0.969 |
| Semantic Segmentation | mIoU↑ | 0.321 | 0.424 | 0.525 | 0.572 | 0.585 | 0.059 | 0.410 | 0.606 |
| | bIoU ↑ | 0.146 | 0.210 | 0.281 | 0.315 | 0.325 | 0.027 | 0.211 | 0.343 |

the average bytes per image with our models. Specifically, we converted the token count to bytes using a rate of 1.5 bytes per token (12 bits). For JPEG and WebP, we used the minimum quality settings to minimize byte size. For JPEG 2000, we verified that a target compression rate of 0.002 yielded an average byte size of 406.3, comparable to our models with 256 tokens. Notably, these traditional image formats performed poorly on the rFID metric compared to neural image tokenizers, including our models.

It is worth mentioning that the PSNR metric shows opposite trends; our models exhibit relatively lower performance compared to image formats. This is due to training objective of our models, which includes GAN loss and perceptual loss, focusing on improving perceptual quality rather than reducing pixel-level distortion. This result aligns with (Blau & Michaeli, 2019), where rate-distortion-perception trade-off is reported. Nevertheless, as we can confirm in Figure 2, our tokenizers exhibit far better perceptual quality compared to these image formats with supporting variable-length tokenization.

To evaluate reconstruction quality with respect to token length, we present the rFID curve in Figure 4. The rFID steadily improves as more tokens are used, and our models demonstrate significantly higher efficiency compared to algorithm-based image compression formats. Notably, our model consistently outperforms ALIT. ElasticTok exhibits superior perceptual quality compared to existing image formats in the low to middle bpp range; however, unlike our model, its performance degrades in the ultra-low to low bpp regime. We also compare the rFID of our models with TiTok variants in Table 3. While our models achieve the lowest rFID at 256 tokens, they do not match TiTok at smaller token counts (32, 64, and 128) when using equivalent model

sizes. We attribute this to the introduction of the Tail Token Drop, which may reduce reconstruction fidelity at small token lengths. Improving perceptual quality in this regime remains an open challenge for future work.

## 4.2. Downstream Tasks

We evaluate the reconstructed images on various computer vision tasks, including image classification, object detection, semantic segmentation, depth estimation, and CLIP embedding reconstruction, to demonstrate the effectiveness of the tokenizer in real-world applications. This evaluation allows us to assess how well the reconstructed images retain critical information, highlighting the balances between compression efficiency and task performance. Table 2 shows the downstream task performance of One-D-Piece-L-256, JPEG and WebP.

### 4.2.1. TASK SETTINGS

**Image Classification.** We use the ImageNet validation split and measure Acc@1 and Acc@5 by comparing classification results on reconstructed images. ConvNeXT (Liu et al., 2022) outputs are used as the ground truth.

**Object Detection.** We use the COCO val2017 (Lin et al., 2014) and employ YOLO11x (Jocher & Qiu, 2024) as the detection model. Performance is evaluated with mAP@0.5:0.95, mAP@0.5, and mAP@0.75.

**Semantic Segmentation.** We evaluate mean Intersection over Union (mIoU) and boundary IoU (bIoU) on the ImageNet-S (Gao et al., 2021) dataset, which provides high-quality semantic segmentation annotations based on the ImageNet-1K. SERE (Gao et al., 2021) is used for the segmentation model.

**Depth Estimation.** We use the ImageNet validation split with Depth Anything (Yang et al., 2024) serving as the ground truth. We evaluate the L1 and L2 error of depth estimations between the reconstructed images and the ground truth.

**CLIP Embedding Reconstruction.** We assess the quality of semantic reconstruction using CLIP embeddings, with the ImageNet validation split and CLIP (Radford et al., 2021) as the ground truth. Cosine similarity is computed between the embeddings of the reconstructed and the original images.

### 4.2.2. RESULTS

As shown in Table 2, One-D-Piece outperforms JPEG across all tasks with only 16 tokens. For tasks where semantic information is crucial, such as CLIP Embedding Reconstruction and Semantic Segmentation, One-D-Piece with 32 or 64 tokens surpasses WebP, achieving a CLIP score of 0.866 versus 0.826 for WebP and an mIoU of 0.424 versus 0.410. In tasks focused on object representation, such as Object Detection and Image Classification, One-D-Piece outperforms WebP with 64 tokens, reaching an mAP@0.5:0.95 of 0.180 versus 0.166 for WebP, and Acc@1 and Acc@5 scores of 0.731 and 0.908, compared to 0.664 and 0.870 for WebP. For Depth Estimation, which requires pixel-level detail, One-D-Piece achieves better L1 and L2 Loss scores at 128 tokens, with values of 1.120 and 3.160, compared to 1.553 and 5.050 for WebP.

Our results show that One-D-Piece uses only 128 tokens, approximately 10% of WebP's byte size per image, yet outperforms WebP across all tasks. Its high compression and preserved quality make it ideal for applications like visual question answering and image or video generation.

### 4.3. Analysis

One-D-Piece demonstrates strong performance in reconstruction quality and adaptability for downstream tasks. We further explore the potential of this novel approach and uncover the contribution of Tail Token Drop and the behavior of and our models.

**Head Tokens Have More Contribution.**

Our Tail Token Drop technique aims to encourage important information to be concentrated at the head of the image token sequence. To verify this hypothesis, we analyze the contribution of each token in the tokenized sequence $\mathbf{q} = [q_1, q_2, \ldots, q_n]$ towards the reconstructed image $\hat{\mathbf{X}}$. Specifically, we perform random replacement for each token $q_i$ in the sequence. Let $\mathbf{q}' = [q_1, \ldots, q_{i-1}, q_i', q_{i+1}, \ldots, q_n]$ be the modified sequence where $q_i$ is replaced by a randomly sampled token $q_i'$. We then reconstruct the image $\hat{\mathbf{X}}'$ from the modified sequence $\mathbf{q}'$. To measure the contribution of each token $q_i$, we compute the L1 error between the origi-

nal reconstruction $\hat{\mathbf{X}}$ and the modified reconstruction $\hat{\mathbf{X}}'$ as $\|\hat{\mathbf{X}}' - \hat{\mathbf{X}}\|_1$. A larger L1 error indicates a greater contribution of the token $q_i$ towards the reconstruction. Furthermore, by retaining the pixel-wise L1 error for each token replacement, we can visualize the spatial regions of the image that are influenced by specific token positions $i$. This allows us to better understand the relationship between token positions in $\mathbf{q}$ and the corresponding pixel areas in $\hat{\mathbf{X}}$. We use ImageNet validation split to examine average behaviors.

The results are shown in Figure 5 (A). As expected, One-D-Piece tokenizers exhibit a strong peak (yellow) at the head of the token sequence, while TiTok models show mostly random peaks. This confirms that our Tail Token Drop approach effectively encourages the model to concentrate important information at the start of the token sequence, resulting in large contributions.

Interestingly, we observe that some middle and later tokens still correspond to specific spatial regions of the image, despite the tokenizer being trained as a purely 1D tokenizer, as shown in Figure 5 (B). This suggests that tokens at certain indices retain a strong connection to the 2D structure of the input image, even under the Tail Token Drop constraint.

Additionally, the tokens at the very end of the sequence show almost no contribution, suggesting they carry little meaningful information and that the latent space is not fully utilized. This indicates the potential for creating a more efficient tokenizer by packing more information into currently less important tokens. This is a future research direction.

**First Token Encodes Global Information.** While our analysis of token contribution confirm that more important information is concentrated in the head tokens, the actual content of these information remains unclear. To qualitatively investigate this, we perform clustering by the first token and present the result.

Figure 6 (A) illustrates ImageNet validation split, clustered by the first token. It can be observed that the first tokens capture the overall similarity between images, indicating that they include global information about the images. Furthermore, we find that replacing the first token leads to corresponding changes in the reconstructed image as shown in Figure 6 (B) . Although this effect diminishes with longer token sequences, significant influence can be seen in the reconstructions with shorter sequences.

**One-D-Piece Tokens are More Semantic.** Our analysis reveals that the Tail Token Drop effectively aggregates global information toward the head of the token sequence, with the leading tokens capturing the overall similarity of the images. To further examine the semantic behavior of One-D-Piece tokens, we conduct linear probing experiment, following the MAE protocol (He et al., 2022), adopted in the TiTok report (Yu et al., 2024b). Specifically, we append a linear

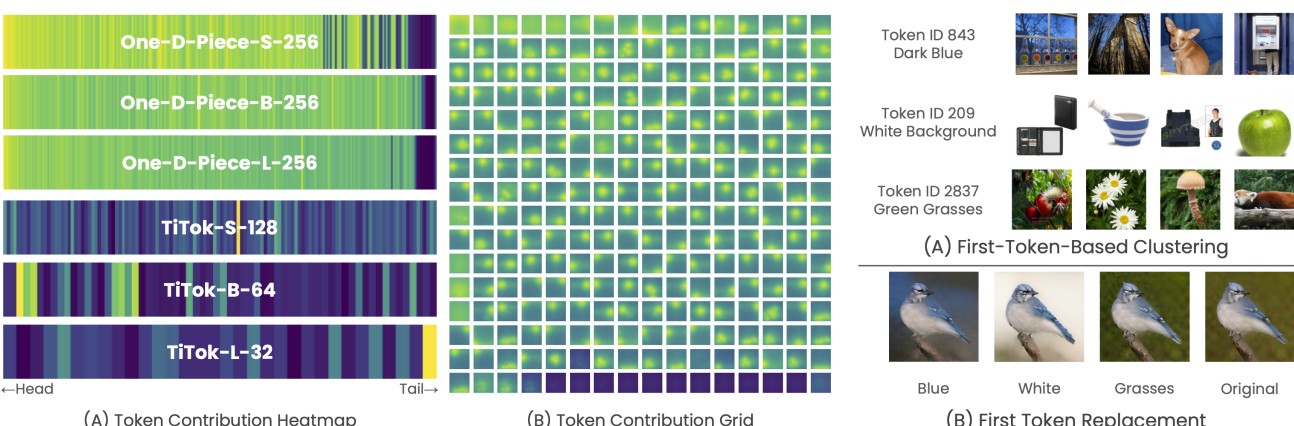

Figure 5: **The Visualization of Our Token Contribution Analysis.** (A) The head tokens capture global information, as indicated by the strong yellow color, while the later tokens show more localized and weaker peaks. (B) Heatmaps of token contributions from the One-D-Piece-L-256 model, displayed in a grid layout for all 256 tokens. Each map highlights the spatial regions to which each token most strongly corresponds. The early tokens capture global features of the entire image, while the mid-to-late tokens respond more strongly to localized, specific regions.

Figure 6: **The Result of First-Token-based Analysis.** (A) The first tokens correspond to global, especially the background information of the image. (B) By replacing the first token to another tokens, we can observe the background changes accordingly, especially when fewer token reconstruction.

classifier for the output of One-D-Piece encoder and train it on the ImageNet-1K classification task. This evaluation measures the linear separability of the encoded representations, indicating how well the latent features capture semantic information.

As shown in Table 3, our model achieves superior linear probing accuracy (LPA) compared to pretrained TiTok models. While our S-sized model exhibits slightly lower LPA than the TiTok model of the same size, our models in the other two sizes significantly outperform the TiTok model. We attribute this high LPA accuracy with fewer tokens to our Tail Token Drop technique, which effectively emulates the benefits of a compact latent space by discarding less informative tail tokens. This result further highlights the effectiveness of the Tail Token Drop approach promoting global information aggregation.

Table 3: **Comparison of rFID and Linear Probing Accuracy (LPA)** between TiTok and One-D-Piece models. For fair comparison, LPA for One-D-Piece is based on the same number of prefix tokens as the equivalent TiTok models.

| Model | rFID↓ | | | LPA↑ |
|---|---|---|---|---|
| | @32 | @64 | @128 | |
| TiTok-S-128 | — | — | 1.70 | 0.349 |
| TiTok-B-64 | — | 1.71 | — | 0.276 |
| TiTok-L-32 | 2.21 | — | — | 0.281 |
| One-D-Piece-S-256 | 7.36 | 3.89 | 1.96 | $0.341_{@128}$ |
| One-D-Piece-B-256 | 4.36 | 2.39 | 1.48 | $0.351_{@64}$ |
| One-D-Piece-L-256 | 3.23 | 2.10 | 1.42 | $0.355_{@32}$ |

**Insights for Autoregressive Image Generation Models.** Building on the analyses above, we trained an autoregressive image generator based on the LlamaGen architecture (Sun et al., 2024). Our experiments reveal several practical insights for improving autoregressive generation.

First, although LlamaGen assumes a raster-ordered 2D tokenizer and employs a 2D variant of Rotary Position Embeddings (RoPE) (Su et al., 2024), this design cannot be directly transferred to 1D tokenizers. We thus compare the standard 1D RoPE with learnable absolute positional embeddings.

Second, because One-D-Piece orders tokens by perceptual importance, errors in the earliest tokens propagate and degrade the whole image. Motivated by this prior, we applied a linearly decaying weight to the cross-entropy loss, assigning greater importance to early tokens.

Figure 7 shows generation quality across Classifier-Free Guidance (CFG) scales. We observe a clear trade-off: RoPE yields lower gFID, while learnable embeddings improve Inception Scores (IS). Increasing the loss weight on head tokens improves both IS and gFID across all CFG scales. While our model does not reach the competitive gFID range of 2–3, leveraging prior knowledge of token importance remains a promising direction for improving autoregressive generation.

We also observe that One-D-Piece integrates well with discrete diffusion-based models, achieving competitive results. Additional experiments using a MaskGIT-based generator are provided in Appendix C.3.

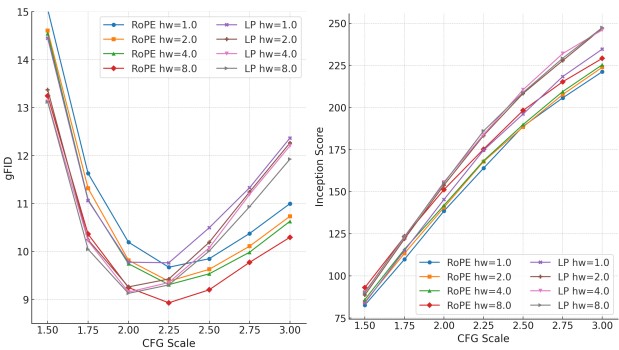

Figure 7: **Generation quality vs. CFG Scale (gFID↓ / IS↑).** Eight settings combine positional embedding (RoPE or learnable, LP) with head-loss weights 1–8. RoPE lowers gFID, LP raises Inception Score, and heavier head-loss weights consistently improve both metrics, most notably near CFG 2–2.5.

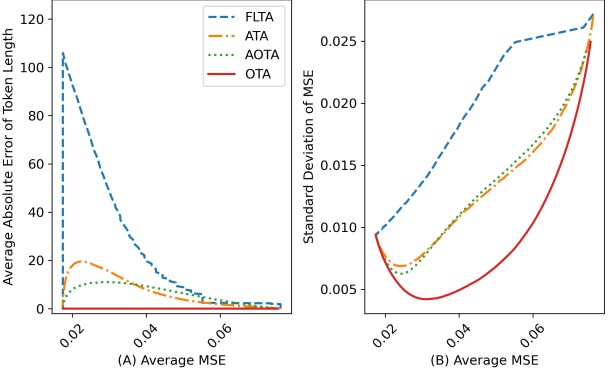

Figure 8: **Comparison of allocation methods for One-D-Piece-S-256.** (A) illustrates the relationship between quality and average error. (B) illustrates the relationship between average quality and standard deviation. ATA achieves competitive performance compared to AOTA.

**Adaptive Token Allocation is Feasible** Since One-D-Piece supports variable-length tokenization, computational efficiency in practice can be improved by assigning an optimal token count to each image. While One-D-Piece supports variable-quality tokenization, determining the optimal token count is challenging. Finding the optimal token count through iterative reconstruction and evaluation involves significant computational costs. To address this, we introduce a model-based Adaptive Token Allocation (ATA) method that efficiently predicts optimal token counts using lightweight neural networks.

In the ATA framework, we predict a 256-dimensional vector for each image using a ResNet-50D model (He et al., 2018) trained to estimate quality metrics (MSE and LPIPS) at each token length. By searching this vector, we efficiently determine the optimal token count for arbitrary quality thresholds. We compare ATA against Fixed-Length Token Allocation (FLTA), Optimal Token Allocation (OTA), and Approximate Optimal Token Allocation (AOTA). FLTA assigns a fixed number of tokens without considering image-specific quality. OTA, in contrast, reconstructs the image at all possible token lengths and selects the minimal length meeting the quality threshold, achieving optimal allocation but at a prohibitively high computational cost. AOTA approximates OTA by evaluating token lengths at 16-token intervals instead of searching over the full range, reducing the computational cost to 1/16 while maintaining near-optimal performance.

To evaluate ATA performance, we adopt two primary metrics: (A) Average Error: Measured as the average absolute error between the estimated and optimal token counts. (B) Standard Deviation of Quality Metrics: Variability in qual-

ity metrics (MSE and LPIPS), where lower values indicate more stable quality control.

Figure 8 shows evaluation results. ATA achieves significantly lower average error and more stable quality control than FLTA. Notably, even compared to AOTA, ATA demonstrates competitive performance across both criteria. Unlike AOTA and OTA, which require high computational cost for token allocation, ATA efficiently estimates the required token count without reconstruction, making it much more computationally efficient. These results indicate that even a simple method like ATA can effectively and dynamically enhance the performance of One-D-Piece. Additional results are presented in Appendix D.

## 5. Conclusion

We introduced One-D-Piece, a discrete image tokenizer enabling quality-controllable tokenization to address the trade-off between compression rate and reconstruction quality in fixed-token methods. Our approach supports dynamic token counts (1–256) using Tail Token Drop, which concentrates critical information at the start of the sequence, allowing high reconstruction quality with fewer tokens and improving perceptual quality at comparable byte sizes. Experiments on ImageNet-1K show that One-D-Piece achieves an rFID of 1.08 with 256 tokens, surpassing prior methods, while excelling in downstream tasks. Our analysis verifies the feasibility of Adaptive Token Allocation and the effectiveness of Tail Token Drop in concentrating key information at the sequence head. These results establish One-D-Piece as an efficient, adaptive tokenizer that sets a new benchmark in compression efficiency and quality, with promising applications in vision-language models and image and video generation.

**Limitations.** Although the architecture supports a higher number of tokens, our evaluation was limited to a maximum of 256. The impact of increasing the token count on reconstruction quality and pixel-level performance remains unexplored. Additionally, we observed that character reconstruction performance is relatively poor. Future work will focus on extending the maximum token count to improve accuracy and quality, as well as developing more advanced training protocols to better preserve important details.

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

# A. Other Related Work

**Neural Image Codec.** Neural Image Codecs and Image Tokenizers share a common motivation for compression, and numerous studies have explored variable-rate compression. However, the representations produced by Neural Image Codecs are often unsuitable for generation, downstream task adaptation, or vision-language integration. Many methods achieve variable-rate compression via rate-conditional autoencoders (Choi et al., 2019; Song et al., 2021; Cui et al., 2021; Duan et al., 2023; Iwai et al., 2024; Jia et al., 2024). While these methods are effective as variable-bitrate compression algorithms, they are impractical as variable-length image tokenizers due to the lack of representational compatibility across different compression rates. In contrast, the One-D-Piece Encoder is rate-agnostic, ensuring token compatibility across different rates, which is an important requirement of image tokenizers.

# B. Training Details

For our experiments, we strictly adhered to the TiTok settings (Yu et al., 2024b) as detailed in Table 7 for both model architecture and training configurations. As the base implementation of TiTok, we utilized the repository bytedance/1d-tokenizer, referencing commit ID `6cf0d6`, following its Apache 2.0 License.

# C. Evaluation Details

## C.1. Reconstruction

**Additional Samples.** We further provide sample reconstructed images, where L-256, B-256, and S-256 are shown in Figure 11, Figure 12, and Figure 13, respectively.

**rFID Values.** We provide the detailed rFID values corresponding to Figure 4 in Table 4. JPEG, JPEG 2000, and WebP format files were generated using OpenCV version 4.10.0.84 as the converter.

## C.2. Downstream Tasks

In addition to the reported result of One-D-Piece-L-256, we further provide results for other variants. The evaluation results for downstream tasks using One-D-Piece-S-256 and B-256 are presented in Table 6. Consistent with our report in the main paper, tasks that rely primarily on semantic information, such as semantic segmentation and CLIP embedding reconstruction, achieve scores that surpass WebP with a smaller number of tokens. Similarly, object detection and image classification tend to show better results with a moderate number of tokens. For the depth estimation task, both models require 128 tokens to outperform WebP.

Table 4: Comparison of image quality against compression quality for JPEG, JPEG 2000, and WebP.

| Method | Quality | Bits per Pixel | Tokens | rFID↓ |
|--------|---------|----------------|--------|-------|
| JPEG | 1 | 0.252 | 1376 | 113.30 |
| | 2 | 0.252 | 1376 | 113.24 |
| | 4 | 0.290 | 1583 | 79.23 |
| | 8 | 0.400 | 2167 | 32.92 |
| | 16 | 0.580 | 3167 | 15.76 |
| | 24 | 0.732 | 3997 | 12.12 |
| JPEG 2000 | 2 | 0.050 | 271 | 299.40 |
| | 4 | 0.097 | 531 | 139.99 |
| | 8 | 0.192 | 1049 | 79.90 |
| | 16 | 0.382 | 2088 | 43.23 |
| | 24 | 0.573 | 3130 | 26.40 |
| | 30 | 0.716 | 3913 | 19.09 |
| WebP | 0 | 0.240 | 1310 | 31.98 |
| | 2 | 0.270 | 1472 | 28.81 |
| | 4 | 0.314 | 1717 | 25.05 |
| | 8 | 0.383 | 2729 | 16.26 |
| | 16 | 0.492 | 2729 | 16.26 |
| | 32 | 0.711 | 3884 | 11.47 |

## C.3. Generation

Our primary evaluations of One-D-Piece focus on reconstruction quality, but generation quality is also an essential metric, particularly for applications like image and video generation models. Since TiTok (Yu et al., 2024b), the base architecture of One-D-Piece, is reported to contribute to high generation quality, our objective is to confirm that the introduction of the Tail Token Drop mechanism does not adversely affect generation quality.

For the evaluation, we train MaskGIT (Esser et al., 2021) for class-conditioned image generation, utilizing One-D-Piece as the image tokenizer, following the protocols utilized in TiTok. We assess the generation FID (gFID) using precomputed statistics from the Ablated Diffusion Model (Dhariwal & Nichol, 2024). As shown in Table 5, our models demonstrate competitive performance compared to TiTok variants. This result confirms that the introduction of Tail Token Drop does not damage the generation quality and highlight the potential of One-D-Piece for image and video generation tasks. Our detailed setting for the generation model is shown in Table 8. Sample generated images are shown in Figure 15.

---

[1] Shanghua Gao, Pan Zhou, Ming-Ming Cheng and Shuicheng Yan. Masked Diffusion Transformer is a Strong Image Synthesizer. in 2023 IEEE/CVF International Conference on Computer Vision (ICCV), Paris, France, 2023, pp. 23107-23116.

Table 5: Generation quality comparison between TiTok and One-D-Piece models, evaluated using gFID (lower is better) and Inception Score (IS, higher is better). The Inception Score for TiTok-S-128 is not reported, as the pretrained MaskGIT with Vision Transformer backbone model for this configuration has not been released.

| Model | TiTok | | | One-D-Piece | | |
|---|---|---|---|---|---|---|
| | S-128 | B-64 | L-32 | S-256 | B-256 | L-256 |
| gFID↓ | 2.50 | 2.48 | 2.77 | 2.67 | 2.70 | **2.35** |
| IS↑ | — | 216.61 | 201.85 | **265.82** | 259.27 | 224.38 |

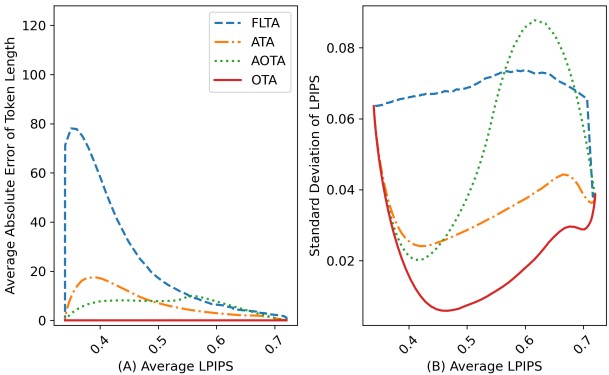

Figure 9: **Comparison of allocation methods for One-D-Piece-S-256 for LPIPS target.** (A) illustrates the relationship between quality and average error. (B) illustrates the relationship between average quality and standard deviation.

### C.4. Inference Speed

While Tail Token Drop reduces the actual token count, it does not decrease the time complexity of the tokenizer. During tokenization, the inference process generates the maximum number of tokens, ensuring that the method introduces no difference on computational cost to the TiTok architecture. However, during detokenization, as fewer tokens are processed, some speed improvements can be observed.

### D. Analysis Details

**Adaptive Token Allocation.** Our approach employs a pretrained ResNet-50D model from timm [2] as the Quality Estimator. Its final linear layer is modified to output 256 values, each corresponding to a potential token count from 1 to 256. An input image is first processed through the One-D-Piece encoder to generate a sequence of 256 tokens. Then, a token count $k$ is uniformly sampled from the range 1 to 256, and the image is reconstructed using only the first $k$ tokens. The reconstruction quality is then evaluated using a chosen metric—whether it be MSE, LPIPS, or CLIP Em-

---

[2] https://huggingface.co/docs/timm/models/resnet-d

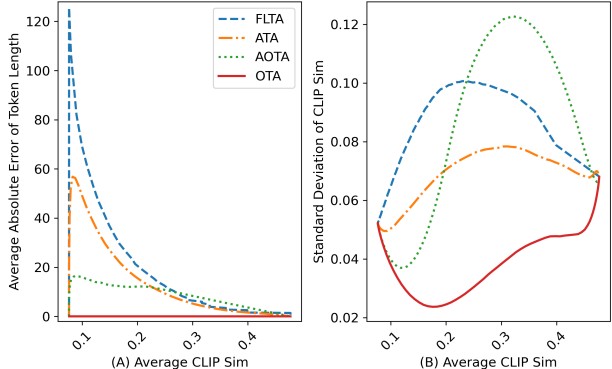

Figure 10: **Comparison of allocation methods for One-D-Piece-S-256 for CLIP embedding similarity target.** (A) illustrates the relationship between quality and average error. (B) illustrates the relationship between average quality and standard deviation.

bedding similarity—and the resulting value quantifies how well the image has been reconstructed with $k$ tokens. This computed quality score is used as the target value for the $k$-th element of the Quality Estimator's output vector. In other words, if the reconstruction using $k$ tokens achieves a particular quality level according to the chosen metric, the Quality Estimator is trained to predict that exact value at its $k$-th position. L1 loss is calculated between the predicted value and the measured quality score, and the entire system is optimized using AdamW optimizer to minimize this loss. Detailed hyper-parameter settings is shown in Table 10. We validate the effectiveness of this method not only with MSE but also with LPIPS and CLIP embedding similarity, demonstrating its versatility in efficiently determining the optimal token count across various quality criteria. The results for LPIPS and CLIP embedding similarity are presented in Figure 9 and Figure 10, respectively.

**Token Contribution Grids.** We provide token contribution grids for all three One-D-Piece variants and three TiTok variants in Figure 16. Note that these visualizations are log-scaled and normalized using the global maximum and minimum values within each grid. As a result, the colors are not directly comparable across different models.

### E. Licenses

The licenses of datasets and models used for training, evaluation, and downstream tasks are described as follows.

**ImageNet-1K** ImageNet-1K comprises 1,281,167 training images, 50,000 validation images, and 100,000 test images, covering a total of 1,000 object classes. We use it under the terms of its access agreement, which permits us-

Table 6: **Evaluation of downstream tasks at different token lengths for One-D-Piece-S-256 and One-D-Piece-B-256.** Green background indicate where One-D-Piece-S-256 surpasses WebP, while yellow background show where One-D-Piece-B-256 surpasses WebP.

| Task | Metrics | One-D-Piece-S-256 | | | | | Image Formats | | Base |
|------|---------|------|------|------|------|------|------|------|------|
| | | @16 | @32 | @64 | @128 | @256 | JPEG | WebP | |
| Object Detection | mAP@0.5:0.95↑ | 0.030 | 0.063 | 0.125 | 0.204 | 0.244 | 0.001 | 0.166 | — |
| | mAP@0.5↑ | 0.062 | 0.112 | 0.197 | 0.300 | 0.349 | 0.001 | 0.217 | — |
| | mAP@0.75↑ | 0.025 | 0.061 | 0.129 | 0.214 | 0.260 | 0.001 | 0.178 | — |
| Depth Estimation | L1 Loss↓ | 2.919 | 2.364 | 1.861 | 1.482 | 1.340 | 3.742 | 1.553 | — |
| | L2 Loss↓ | 16.006 | 11.478 | 7.590 | 5.024 | 4.171 | 24.097 | 5.050 | — |
| CLIP Emb Reconstruction | Cos Sim↑ | 0.779 | 0.832 | 0.879 | 0.914 | 0.926 | 0.610 | 0.826 | — |
| Image Classification | Acc@1↑ | 0.378 | 0.535 | 0.659 | 0.738 | 0.759 | 0.284 | 0.664 | 0.841 |
| | Acc@5↑ | 0.613 | 0.769 | 0.864 | 0.915 | 0.928 | 0.479 | 0.870 | 0.969 |
| Semantic Segmentation | mIoU↑ | 0.2016 | 0.329 | 0.438 | 0.518 | 0.540 | 0.059 | 0.410 | 0.606 |
| | bIoU↑ | 0.084 | 0.154 | 0.223 | 0.278 | 0.295 | 0.027 | 0.211 | 0.343 |

| Task | Metrics | One-D-Piece-B-256 | | | | | Image Formats | | Base |
|------|---------|------|------|------|------|------|------|------|------|
| | | @16 | @32 | @64 | @128 | @256 | JPEG | WebP | |
| Object Detection | mAP@0.5:0.95↑ | 0.038 | 0.080 | 0.148 | 0.228 | 0.277 | 0.001 | 0.166 | — |
| | mAP@0.5↑ | 0.076 | 0.140 | 0.234 | 0.337 | 0.391 | 0.001 | 0.217 | — |
| | mAP@0.75↑ | 0.034 | 0.079 | 0.152 | 0.235 | 0.292 | 0.001 | 0.178 | — |
| Depth Estimation | L1 Loss↓ | 2.709 | 2.182 | 1.723 | 1.377 | 1.214 | 3.742 | 1.553 | — |
| | L2 Loss↓ | 14.404 | 10.095 | 6.682 | 4.411 | 3.491 | 24.097 | 5.050 | — |
| CLIP Emb Reconstruction | Cos Sim↑ | 0.798 | 0.849 | 0.891 | 0.920 | 0.934 | 0.610 | 0.826 | — |
| Image Classification | Acc@1↑ | 0.441 | 0.586 | 0.697 | 0.756 | 0.776 | 0.284 | 0.664 | 0.841 |
| | Acc@5↑ | 0.672 | 0.806 | 0.890 | 0.926 | 0.938 | 0.479 | 0.870 | 0.969 |
| Semantic Segmentation | mIoU↑ | 0.250 | 0.372 | 0.480 | 0.536 | 0.562 | 0.059 | 0.410 | 0.606 |
| | bIoU↑ | 0.108 | 0.180 | 0.250 | 0.291 | 0.309 | 0.027 | 0.211 | 0.343 |

age for non-commercial research and educational purposes.

**ImageNet-S** ImageNet-S provides high-quality semantic segmentation annotations for robust evaluation, based on 12,419 validation images and 26,423 test images sourced from ImageNet. The dataset focuses on 919 categories, excluding unsegmentable ones such as "bookshop." Usage of this dataset adheres to the ImageNet licensing terms.

**COCO** COCO val2017 consists of 5,000 images spanning 80 categories, including a wide range of annotated objects such as people, animals, vehicles, and furniture. We follow the terms of use of COCO and the Flickr Terms of Use for images in the COCO dataset.

**Ultralytics YOLO11** We use Ultralytics YOLO11x for object detection. Ultralytics YOLO11 is released under the GNU Affero General Public License v3.0 (AGPL-3.0), which permits the use of the model for research purposes.

Table 7: **Hyperparameters for One-D-Piece models**. These hyperparameters are fully following TiTok settings.

| Item | Value |
|------|-------|
| **Model** | |
| Codebook Size | 4,096 |
| Token Size | 12 |
| Model Size | ViT small / base / large |
| Patch Size | 16 |
| Latent Tokens | 256 |
| **Training** | |
| Stage1 Epochs | 100 |
| Stage2 Epochs | 200 |
| Stage1 Batch Size | 1024 |
| Stage2 Batch Size | 512 |
| Dataset | ImageNet-1K |
| Augmentation | Random Crop / Flip |
| **Losses** | |
| **Stage1** | |
| Pretrained Tokenizer | MaskGIT tokenizer [Link] |
| Target Codebook Size | 1024 |
| Reconstruction Weight | 1.0 |
| Quantizer Weight | 1.0 |
| **Stage2** | |
| Discriminator Weight | 0.01 |
| Perceptual Loss Model | ConvNeXT-Small [Link] |
| Perceptual Loss Weight | 0.1 |
| Reconstruction Weight | 1.0 |
| Commitment Loss Weight | 0.25 |
| Codebook Loss Weight | 1.0 |
| **Optimizer** | |
| Optimizer | AdamW |
| Learning Rate | 1e-4 |
| Beta1 | 0.9 |
| Beta2 | 0.99 |
| Weight Decay | 1e-4 |
| Epsilon | 1e-8 |
| **Scheduler** | |
| Scheduler Type | Cosine |
| Warmup Steps | 10,000 |
| End Learning Rate | 1e-5 |

Table 8: **Hyperparameters for MaskGIT models**. These hyperparameters are fully following TiTok settings.

| Item | Value | | |
|------|-------|-------|-------|
| | **S-256** | **B-256** | **L-256** |
| **Model** | | | |
| Architecture | | MaskGIT | |
| Hidden Dim | | 768 | |
| Hidden Layers | | 24 | |
| Attention Heads | | 16 | |
| Dropout Rate | | 0.1 | |
| Class Label Drop | | 0.1 | |
| Class Count | | 1000 | |
| Latent Tokens | | 256 | |
| **Training** | | | |
| Epochs | | 900 | |
| Batch Size | | 2048 | |
| Dataset | | ImageNet-1K | |
| Augmentation | | Random Flip | |
| **Losses** | | | |
| Loss Function | | CrossEntropy | |
| Label Smoothing | | 0.1 | |
| Unmasked Token Loss | | 0.1 | |
| **Optimizer** | | | |
| Optimizer | | AdamW | |
| Learning Rate | | 2e-4 | |
| Beta1 | | 0.9 | |
| Beta2 | | 0.96 | |
| Weight Decay | | 0.03 | |
| **Scheduler** | | | |
| Scheduler Type | | Cosine | |
| Warmup Steps | | 10,000 | |
| End Learning Rate | | 1e-5 | |
| **Decoding** | | | |
| Steps | 16 | 16 | 16 |
| Temperature | 3.0 | 2.5 | 3.0 |
| Guidance Decay | | Power Cosine [1] | |
| Guidance Scale | 12.5 | 8.5 | 5.5 |

Table 9: **Hyperparameters for LlamaGen models**.

| Item | Value |
|---|---|
| **Model** | |
| Architecture | LlamaGen-L |
| Hidden Dim | 1024 |
| Hidden Layers | 24 |
| Attention Heads | 16 |
| Dropout Rate | 0.0 |
| Class Label Drop | 0.1 |
| Class Count | 1000 |
| Latent Tokens | 256 |
| **Training** | |
| Epochs | 150 |
| Batch Size | 2048 |
| Dataset | ImageNet-1K |
| Augmentation | Random Flip |
| **Losses** | |
| Loss Function | CrossEntropy |
| Label Smoothing | 0.1 |
| **Optimizer** | |
| Optimizer | AdamW |
| Learning Rate | 1e-4 |
| Beta1 | 0.9 |
| Beta2 | 0.95 |
| Weight Decay | 0.05 |
| **Scheduler** | |
| Scheduler Type | Cosine |
| Warmup Steps | 10,000 |
| End Learning Rate | 1e-5 |
| **Decoding** | |
| Temperature | 1.0 |
| Guidance Scale | 1.5-3.5 |
| Top-P | N/A |
| Top-K | N/A |

Table 10: **Hyperparameters for Quality Estimator Training**.

| Item | Value |
|---|---|
| **Training** | |
| Epochs | 10 |
| Batch Size | 64 |
| Dataset | ImageNet-1K |
| Augmentation | None |
| **Optimizer** | |
| Optimizer | AdamW |
| Learning Rate | 1e-4 |
| Beta1 | 0.9 |
| Beta2 | 0.99 |
| Weight Decay | 0.01 |
| Epsilon | 1e-8 |
| **Scheduler** | |
| Scheduler Type | Cosine |
| Warmup Steps | 0 |
| End Learning Rate | 0 |

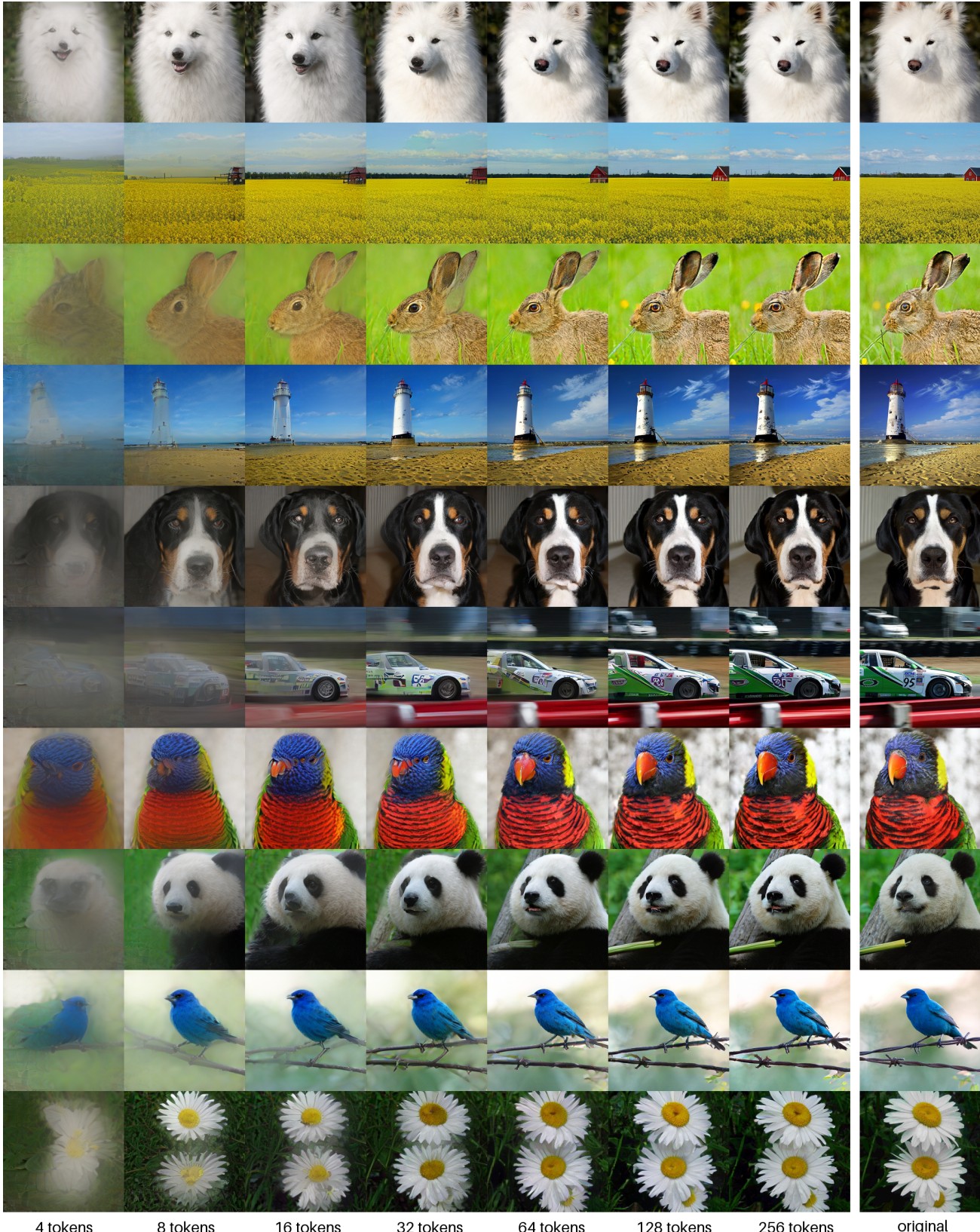

4 tokens  8 tokens  16 tokens  32 tokens  64 tokens  128 tokens  256 tokens  original

Figure 11: **Visual comparison of reconstructed images with One-D-Piece-L-256.**

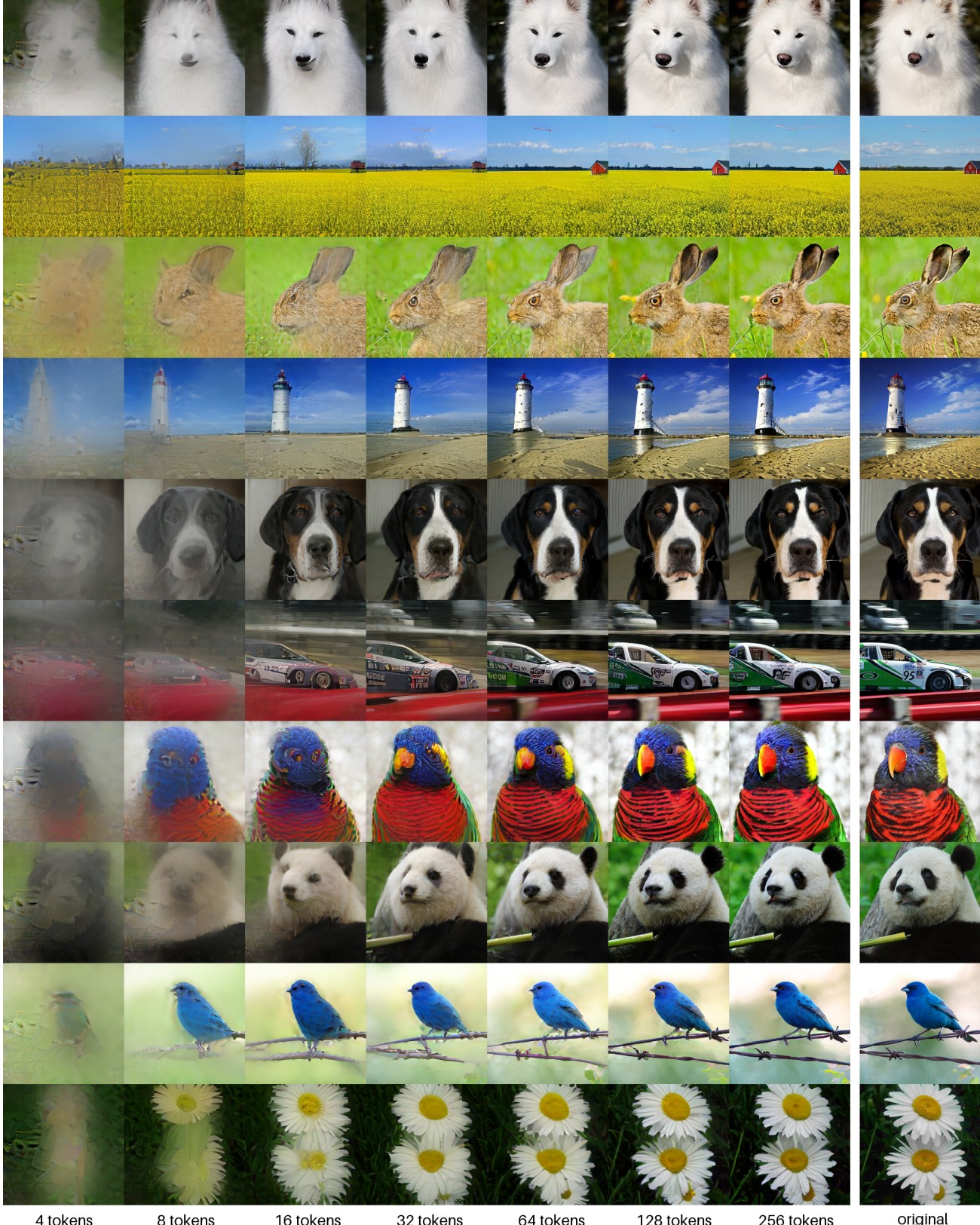

4 tokens    8 tokens    16 tokens    32 tokens    64 tokens    128 tokens    256 tokens    original

Figure 12: **Visual comparison of reconstructed images with One-D-Piece-B-256**.

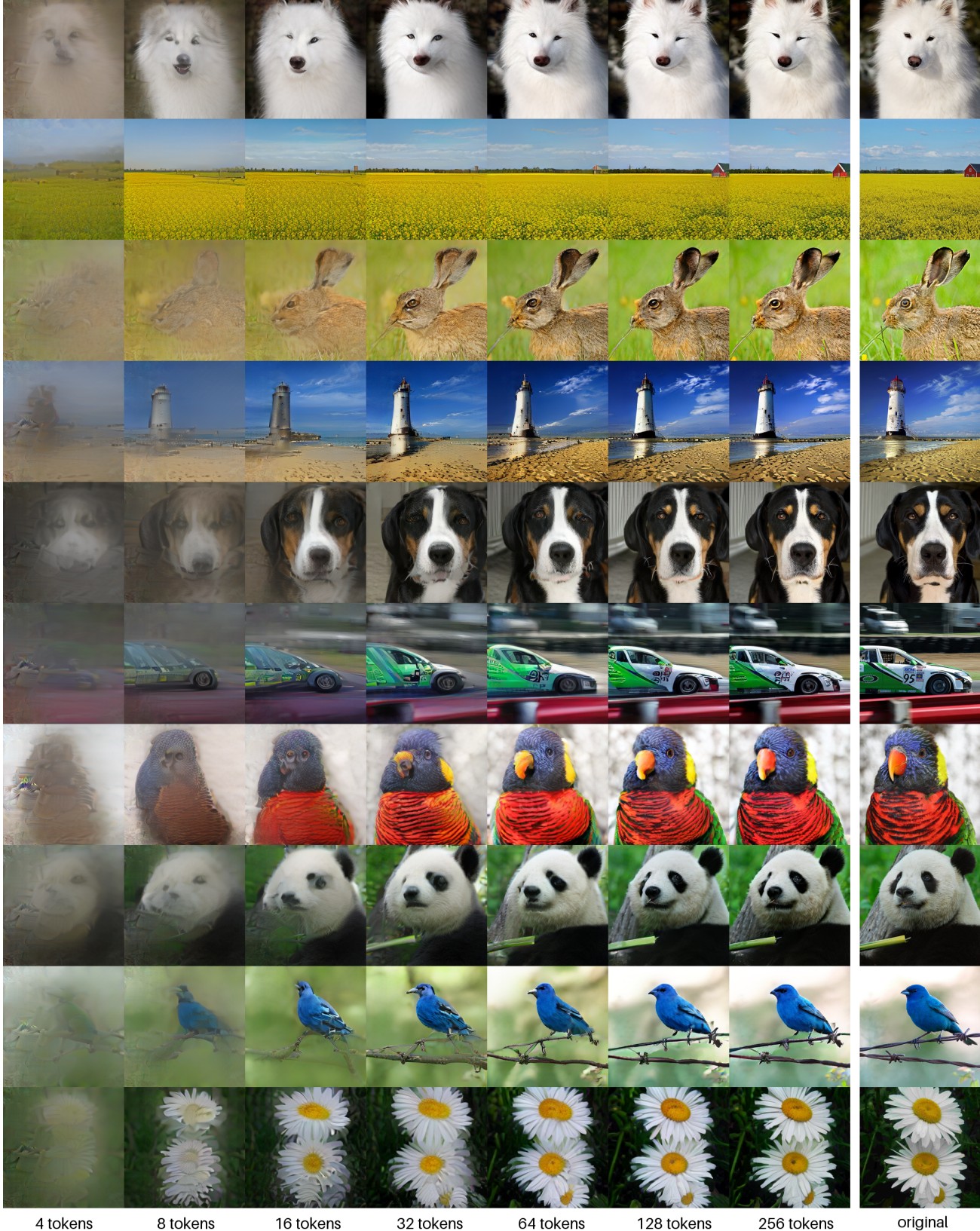

4 tokens    8 tokens    16 tokens    32 tokens    64 tokens    128 tokens    256 tokens    original

Figure 13: **Visual comparison of reconstructed images with One-D-Piece-S-256.**

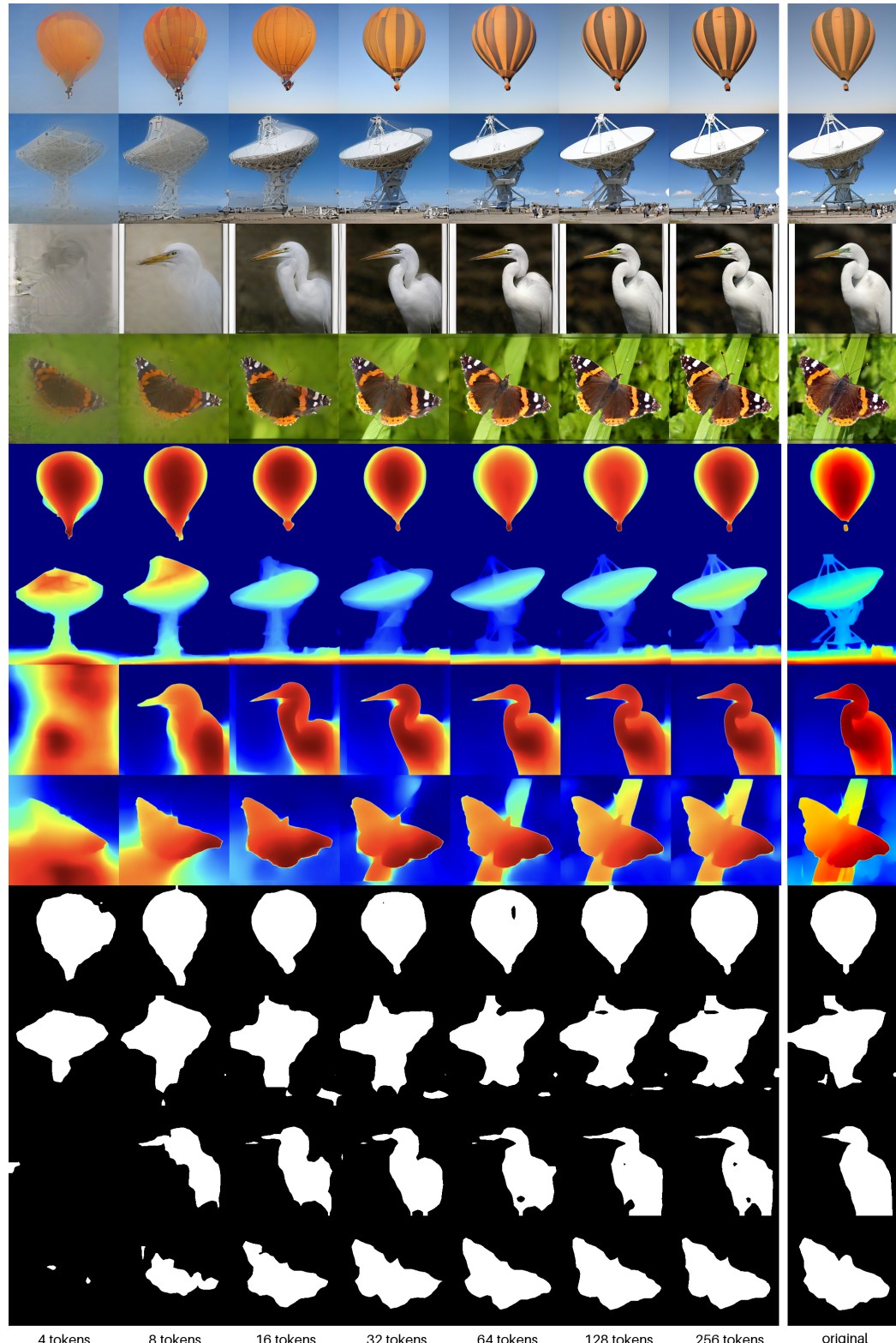

Figure 14: **Results of depth estimation and semantic segmentation on reconstructed images with One-D-Piece-L-256**. With an increase in token count, these results approach those of the original images.

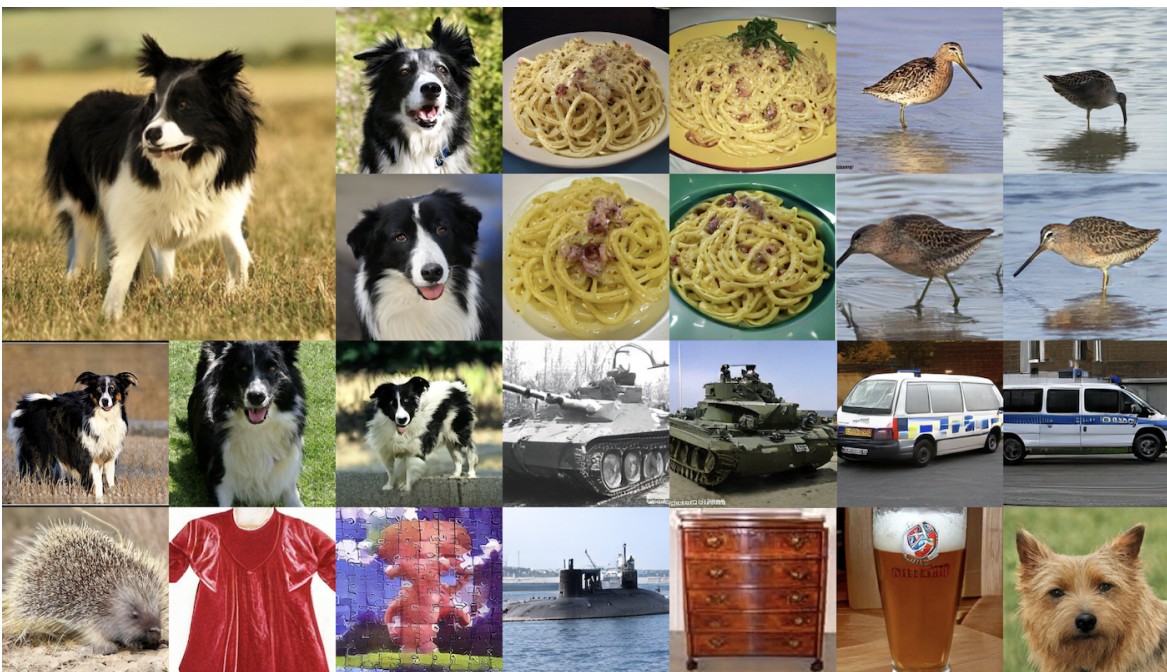

Figure 15: Images generated by MaskGIT with One-D-Piece-L-256 with random classes.

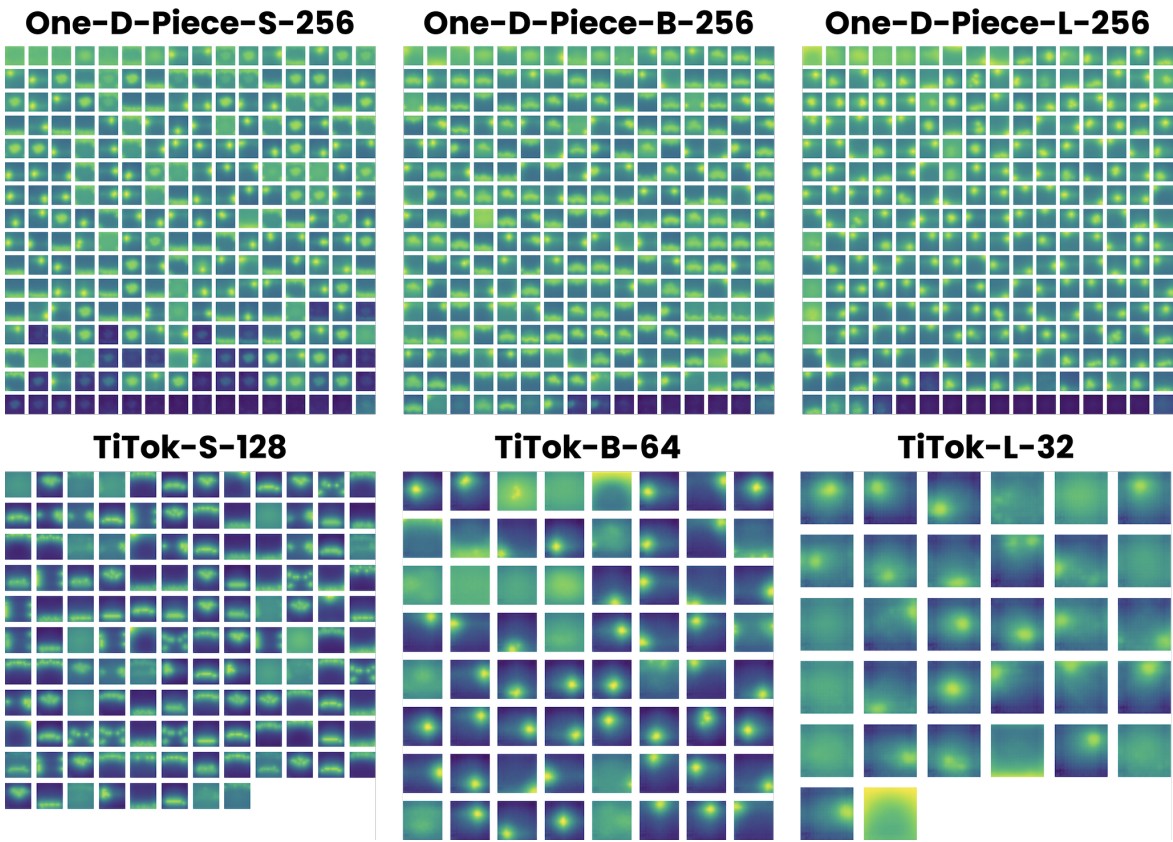

Figure 16: Token Contribution Grid for all variants of One-D-Piece and TiTok. Our One-D-Piece models demonstrate a clear concentration of global information at the head of the token sequence, whereas the TiTok models distribute such tokens more randomly.

