# OpenReview forum: "One-D-Piece: Image Tokenizer Meets Quality-Controllable Compression"
_ICML.cc/2025/Workshop/TokShop — TokShop_

### Official Review · Reviewer_QHoo · 2025-06-08

**Rating:** 7
**Confidence:** 4

**Review:**

This paper presents One-D-Piece, a novel 1D image tokenizer that employs a simple yet effective regularization mechanism called "Tail Token Drop" to enable variable-length tokenization. The proposed approach addresses a key limitation in existing image tokenizers, which typically produce fixed-length token sequences regardless of image complexity.

Pros:

The empirical analysis provides compelling insights into the learned token representations. Particularly noteworthy are the findings that: (1) the first token encodes global image information, as demonstrated through clustering analysis, and (2) the tokenizer learns more semantic representations, evidenced by superior linear probing accuracy compared to baseline methods. These observations suggest that Tail Token Drop not only improves compression efficiency but also enhances the semantic meaningfulness of the learned representations.

Cons & questions:

While the approach looks promising, the paper could benefit from discussing computational overhead during training and potential limitations at extremely low token counts. Additionally, comparison with more recent variable-rate methods would strengthen the evaluation.

Overall:

The paper makes a solid contribution to image tokenization with a simple, effective method supported by thorough empirical analysis and meaningful insights into token behavior.RetryClaude can make mistakes. Please double-check responses.

---

### Official Review · Reviewer_QsaN · 2025-06-08
**Variable length image tokenizer with strong practical utility, though some tradeoffs need further analysis**

**Rating:** 8
**Confidence:** 4

**Review:**

The authors introduce a new discrete image tokenizer for variable length tokenziation and compression that can be controlled based on the quality requirements of the user.

The existing image tokenization methods require a large number of fixed tokens. But one-d-piece allows for variable token lengths ranging from 1 to 256 tokens, based on the quality requirements from the compression. TiTok had introduced a similar approach where they compressed images to only 32 tokens, but the tradeoff between quality and compression efficiency was missing in that approach. The authors hence design a mechanism called “Tail token drop” which allows important image information to be focused more in the head of the token sequence. This enables flexible compression rates, by controlling the the head tokens retained while reconstruction of the image.

Built on the TiTok architecture, they train their model using a 2-stage method, using cross-entropy and reconstruction loss. The authors have shown that this technique achieves significantly better perceptual quality and compression efficiency than traditional formats like JPEG and WebP. The authors also extended their analysis to show this approaches advantages of using far fewer tokens or byte size  in downstream tasks including image classification, object detection, semantic segmentation, etc.

They also validate the effectiveness of the the tail token drop in image generation and adaptive token allocation.

Strengths:
1. Variable length tokenization: Novel approach which addressed the limitation of current image tokenizers, by supporting variable length tokenization allowing flexibility in choosing the compression based on the quality/efficiency requirements.
2. Tail token drop regularization: Inspired from tail-drop regularization (Koike-Akino & Wang, 2020), in the context of image compression, is a simple yet effective approach, which enables the model to accumulate critical information towards the beginning of the sequence length. This enables the paper to present a flexible adjusted sequence length.
3. Significant improvement over traditional compression techinqiues: Better perceptual quality (lower rFID scores) compared to both existing neural image tokenizers (VQGAN, MaskGIT, Open-MAGVIT2) and image formats like JPEG, JPEG 2000 and WebP. The results though show lower PSNR, which is attributed to the objective of the model (which focuses more on perceptual quality than reducing per-pixel distortion), but it clearly shows far better perceptual quality than other techniques.
4. Practical Utility: The paper has a clear use cases for downstream tasks, such as CLIP embedding reconstruction, semantic segmentation, object detection, image classification, etc. Also they demonstrate feasibility of adaptive token allocation, which would be useful in real application, by predicting optimal token count with iterative reconstructions.

Weaknesses:
1. Perceptual Quality at smaller token counts: This approach doesn’t perform better than TiTok’s at smaller token count (32, 64 and 128), using similar model sizes, which can be attributed to the tail token drop. The authors could have done more quantitative analysis to explore this effect more in detail (for ex, reconstruction error with and without the drop at fixed lengths)
2. Character reconstruction: As already stated by the author, reconstruction performance for characters are poor using this approach.
3. More exploration on Token Tail drop schedules: It would have been more helpful if there are more exploration on different sampling distributions or dropout schedules, to quantify the effect on models quality or convergence.

Justification for rating:

Why not higher: Tail token drop needs more analysis like ablation studies, and reconstruction quality at low token count still remains a limitation.
Why not lower: Well executed, novel, shows strong empirical results, practical and broad applicability.

---

### Official Review · Reviewer_9dFE · 2025-06-10
**Paper proposes variable length 1D tokenization of images using "Tail Token Drop", reconstruction metrics look much better compared to baselines, impact on downstream tasks is also studied. Some more baseline would have been better and there's existing work that proposes almost identical method  which should have been covered. Overall, it's a good work that benefit discussion in Tokenization focused audiance.**

**Rating:** 7
**Confidence:** 3

**Review:**

The paper introduces "One-D-Piece" a variable length 1D tokenizer for images. Authors propose "Tail Token Drop" technique where tail tokens are dropped randomly during training which induces an ordering among the 1D tokens with more important tokens concentrating at the head. This allows variable length 1D tokenization. The reconstruction measured by FID is better than all benchmarks at higher Token Count (256). However,  TiTok does better at lower token count. Authors also show downstream task performance using reconstructed images is competitive compared to oracle performance.

Strenghts:
1. Authors propose a neat idea to enable variable length 1D tokenization. Variable length tokenization has one key advantage that the same tokenizer can be used with different token counts unlike TiTok where a new tokenizer needs to be trained for a specific token count.
2. The reconstruction metrics beats TiTok at higher token counts showing promise in the approach.

Weakness
1. For impact on downstream tasks, it's unclear why TiTOK like tokenizers aren't used as a baseline.

NOTE: There's recent work, FlexTok (https://arxiv.org/pdf/2502.13967), accepted at ICML25 which proposes exactly the same idea of dropping tail tokens to create variable length tokenizer. I leave it up to the organizers to decide if it's considered concurrent work or not.

---

### Decision · Program_Chairs · 2025-06-10

Accept